# Role of Annexin A1 Secreted by Neutrophils in Melanoma Metastasis

**DOI:** 10.3390/cells12030425

**Published:** 2023-01-27

**Authors:** Silvana Sandri, Cristina Bichels Hebeda, Milena Fronza Broering, Marina de Paula Silva, Luciana Facure Moredo, Milton José de Barros e Silva, André Sapata Molina, Clóvis Antônio Lopes Pinto, João Pedreira Duprat Neto, Chris P. Reutelingsperger, Cristiane Damas Gil, Sandra Helena Poliselli Farsky

**Affiliations:** 1Department of Clinical and Toxicological Analyses, School of Pharmaceutical Sciences, University of Sao Paulo, São Paulo 05508-000, SP, Brazil; 2NPCMed—Núcleo de Pesquisa em Ciências Médicas, Centro Universitário para o Desenvolvimento do Alto Vale do Itajaí—UNIDAVI, Rio do Sul 89160-932, SC, Brazil; 3Center for Stem Cells & Regenerative Medicine, Kings College London, London WC2R 2LS, UK; 4Skin Cancer Department, A.C. Camargo Cancer Center, São Paulo 01509-010, SP, Brazil; 5Cardiovascular Research Institute Maastricht, Maastricht University Medical Center, Maastricht University, 6211 LK Maastricht, The Netherlands; 6Department of Morphology and Genetics, Universidade Federal de São Paulo (UNIFESP), São Paulo 04023-900, SP, Brazil

**Keywords:** neutrophil-depleted mice, melanoma patients, FPR antagonists, B16F10 cells, neutrophil–lymphocyte ratio (NLR)

## Abstract

Annexin A1 (AnxA1) is highly secreted by neutrophils and binds to formyl peptide receptors (FPRs) to trigger anti-inflammatory effects and efferocytosis. AnxA1 is also expressed in the tumor microenvironment, being mainly attributed to cancer cells. As recruited neutrophils are player cells at the tumor sites, the role of neutrophil-derived AnxA1 in lung melanoma metastasis was investigated here. Melanoma cells and neutrophils expressing AnxA1 were detected in biopsies from primary melanoma patients, which also presented higher levels of serum AnxA1 and augmented neutrophil–lymphocyte ratio (NLR) in the blood. Lung melanoma metastatic mice (C57BL/6; i.v. injected B16F10 cells) showed neutrophilia, elevated AnxA1 serum levels, and higher labeling for AnxA1 in neutrophils than in tumor cells at the lungs with metastasis. Peritoneal neutrophils collected from naïve mice were co-cultured with B16F10 cells or employed to obtain neutrophil-conditioned medium (NCM; 18 h incubation). B16F10 cells co-cultured with neutrophils or with NCM presented higher invasion, which was abolished if B16F10 cells were previously incubated with FPR antagonists or co-cultured with AnxA1 knockout (AnxA1^-/-^) neutrophils. The depletion of peripheral neutrophils during lung melanoma metastasis development (anti-Gr1; i.p. every 48 h for 21 days) reduced the number of metastases and AnxA1 serum levels in mice. Our findings show that AnxA1 secreted by neutrophils favors melanoma metastasis evolution via FPR pathways, addressing AnxA1 as a potential biomarker for the detection or progression of melanoma.

## 1. Introduction

The tumor microenvironment consists of a heterogeneous population of cancer cells and a variety of other cells including the resident and infiltrating host cells [1]. Infiltrating immune cells in the microenvironment distinctly influences the tumor progression. While T-cell-mediated anti-tumor immune response correlates with favorable disease outcomes and is the basis of cancer immunotherapy [2,3,4,5], the myeloid cells act as antigen-presenting cells to promote anti-tumor T-cell responses at the initial phases of tumorigenesis; however, lately, they have been effectors of the tumor progression [6,7,8].

Neutrophils are myeloid cells constantly produced in the bone marrow and released into the blood. During the host defense against aggressions, this pattern is exacerbated, leading to different neutrophil phenotypes with different half-lives and functions [9,10]. Neutrophilia is also detected in cancer patients, especially those with advanced-stage disease [11], and a high neutrophil-lymphocyte ratio (NLR) has been considered a biomarker of poor clinical outcomes in many types of cancer [12,13,14,15]. Neutrophils also represent a significant proportion of immune cells infiltrating in many types of cancer [16,17,18]. At the tumor site, tumor-associated neutrophils (TAN) are plastic cells that adapt to different microenvironments. In the early tumorigenesis phase, the N1 TAN phenotype stimulates antitumor immune response; however, under the continuous action of chemical mediators released in the tumor microenvironment, TAN changes to a N2 immunosuppressive phenotype and stimulates motility, migration, and invasion of tumor cells [19,20].

In melanoma, the most aggressive type of skin cancer, a body of research has emerged employing NLR in the prognostic of both localized and metastatic melanoma [21,22,23,24,25], as in immunotherapy monitoring in melanoma, being high in NLR was associated with poor outcomes [22,26]. It has been pointed out that circulating neutrophils stimulate angiogenesis and enhanced melanoma cell migration toward blood endothelial cells [27], allowing tumor cell invasion; TLR-4 activation in peripheral neutrophils promotes the metastatic spread of melanoma [28]; and neutrophils activated by complement-membrane-attack-complex release neutrophil extracellular traps (NETs), which open the endothelial barrier favoring melanoma cells to reach into the circulation and their systemic spread [29]. Furthermore, circulating neutrophils migrate to the tumor by actions of chemotactic mediators produced in the lung melanoma microenvironment [30,31,32], and immature neutrophils at the tumor-bearing lungs display pro-tumoral effects, such as the enhanced frequency of ROS-producing cells [33] and NET formation [34]; conversely, neutrophils expressing lower amounts of Wip-1, a negative regulator of p53, trigger anti-tumor effects [35], and high infiltration of phagocyting neutrophils evoked by intratumoral injection of vaccine-stimulating biodegradable polysaccharide favors the maturation of dendritic cells and the generation of immune memory [36].

Azurophil granules of neutrophils stock the anti-inflammatory protein Annexin A1 (AnxA1), and in inflammatory conditions, newly synthesized and AnxA1 stored in the neutrophil granules are released into the microenvironment, mainly in microvesicle contents [37]. Secreted AnxA1 is phosphorylated in a calcium-dependent manner and binds to G-protein-coupled receptors named formyl peptide receptors—FPR1 and FPR2—to downstream different intracellular signaling pathways [38,39]. The neutrophil AnxA1 actions are associated with the blockade of leukocyte migration activated at the early inflammation, efferocytosis, and tissue repair [40,41,42]. Moreover, a lower frequency of circulating neutrophil-AnxA1^+^ is associated with a reduction of the plasma levels of the protein, which is found in systemic inflammatory diseases [43], and impaired AnxA1 expression in neutrophils at the site of lesion exacerbates and lengthens the course of inflammation [44].

Many types of cancer cells also express AnxA1, and its effects are associated with cancer development by inducing proliferation, angiogenesis, stemness, and cell invasion in FPR-dependent pathways [45]. Previous studies demonstrated that melanoma cells expressed AnxA1 into the microenvironment, which is associated with angiogenesis, tumor cell invasiveness, and growth [46,47,48]. *In vitro* studies corroborating the expression of AnxA1 by melanoma cells is related to invasiveness behavior, depending, at least in part, on metalloproteinase-2 (MMP-2) expression [48]. The fine-tuned understanding of neutrophils on solid tumors is nowadays a remarkable goal to clarify the mechanisms of tumor progression and to point out a target for treatment. Hence, we here depicted the role of AnxA1 secreted by neutrophils in lung melanoma mice and melanoma patients and unveiled the fact that AnxA1 derived from the blood or tumor-site neutrophils displays pro-tumor invasiveness during lung melanoma development.

## 2. Materials and Methods

### 2.1. Patients

Eighteen-year-old or older patients with histological confirmation of nevus or melanoma by a pathologist were eligible for enrollment in the study. The patients recruited at A.C. Camargo Cancer Center, São Paulo, Brazil, donated blood (8 mL) and authorized access to biopsies for posterior analysis. Healthy donors were included as controls. The study was conducted under the Declaration of Helsinki and approved by the Ethics in Research Committee of the University of São Paulo and A.C. Camargo Cancer Center (CAAE 277951120.0000.0067 and 277951120.4.3001.5432, respectively). In total, 16 patients were included, and sample characteristics are depicted in Table 1. 

### 2.2. Animals

Female C57BL/6 mice wild-type (AnxA1^+/+^**)** or AnxA1-knockout mice (AnxA1^-/-^) (25–30 g; 3–5 per group) were provided by the Central Animal House of the School of Pharmaceutical Sciences and the Chemistry Institute of the University of São Paulo. Mice were housed in polycarbonate cages (four animals per cage; Tecniplast, Buguggiate, Italy) at room temperature (22 °C ± 0.1 °C) and humidity (50% ± 10%) with a 12 h light/dark cycle, receiving standard food and water ad libitum. Animals were anesthetized with a combination of ketamine/xylazine solution (20:2 mg/kg, intraperitoneal (i.p.); xylazine hydrochloride—Ceva Santé Animale; ketamine—Syntec do Brasil Ltda) before each experimental procedure. All procedures were approved by the Institutional Animal Care and Use Committee (IACUC) at the School of Pharmaceutical (CEUA FCF/USP 583).

### 2.3. Melanoma Cell Culture

The B16F10 malignant melanoma cell line was obtained from the Banco Células do Rio de Janeiro (BCRJ), Rio de Janeiro, Rio de Janeiro, Brazil. Cells were grown in culture dishes for 5–10 passages in Dulbecco’s modified Eagle’s medium (DMEM; #12100046 Gibco, Carlsbad, CA, USA) supplemented with 10% heat-inactivated fetal bovine serum (FBS; #2024-06 Gibco) and 100 U/mL penicillin and 100 μg/mL streptomycin solution (#15140-122 Gibco), maintained at 37 °C with 5% CO_2_ atmosphere. Reverse transcription polymerase chain reaction (RT-PCR) was performed routinely in the laboratory to check the mycoplasma cell contamination. AnxA1 and FPR expression in melanoma cells were evaluated by flow cytometer as described below.

### 2.4. Collection of Plasma and Isolation of Blood Human Neutrophils

An aliquot of fresh and heparinized venous blood from patients and control donors was used to total leukocyte and smear blood count. The count of total leukocytes was performed using a Neubauer chamber, and smear blood was stained with Giemsa to circulating leukocyte differentiation. The NLR was calculated by the ratio between the number of circulating neutrophils and lymphocytes counted. Plasma was separated by centrifugation, recovered, and stored at −80 °C until AnxA1 serum levels measurement. Afterward, erythrocytes and leukocytes pellet were diluted in 0.9% NaCl, and neutrophil isolation was carried out by density gradient centrifugation, as described previously [49]. Isolated neutrophils were fixed and analyzed by flow cytometer as described in the following item.

### 2.5. Collection and Culture of Mice Neutrophils

Neutrophils were obtained from wild-type and AnxA1^-/-^ mice, 4 h after intraperitoneal injection of 3 mL 1% oyster glycogen Type II solution (#G8751 Sigma-Aldrich, Saint Louis, MO, USA), previously prepared in phosphate-buffered saline (PBS) [50]. The animals were anesthetized, and the cells were collected by rinsing the abdominal cavity with 10 mL of PBS. To macrophage adherence, peritoneal cells were incubated for 2 h in DMEM supplemented with 10% FBS, 100 U/mL penicillin, and 100 μg/mL streptomycin. Afterward, non-adhered neutrophils were recovered and counted in the Neubauer chamber. Then, neutrophils (1 × 10^6^) were seeded in a 96-well plate (Corning, New York, NY, USA) and cultured in DMEM supplemented with 10% FBS for 18 h. After culture, the cell-free supernatant was recovered as Neutrophil Conditioned Medium (NCM), filtered through a 45 μm filter (Corning), stored at –80 °C, and used according to assays. The purity of the peritoneal cell population and viability were analyzed by flow cytometry.

### 2.6. Induction of Melanoma Lung Metastasis in Mice

Cultured B16F10 cells were harvested, washed, and resuspended to give the appropriate concentration in serum-free DMEM. B16F10 cells suspension (5 × 10^5^ cells/100 μL) was injected into the mice via the tail vein [51]. The animals were maintained under observation, and after 21 days of tumor cell injection, the mice were submitted to euthanasia by overexposure to nasal anesthesia (isoflurane; 2-chloro-2-(difluoromethoxy)-1,1,1-trifluoro-ethane; Cristália, Brazil). For neutrophil depletion, 150 μg anti-Gr1 (16-5931-95; clone RB6-8C5 Bioscience/Invitrogen, Waltham, MA, USA) or rat IgG2b kappa isotype (#14-4031-85 Bioscience/Invitrogen) were injected 24 h before melanoma cell inoculation. The antibodies were injected every 48 h for up to 21 days. The blood was recovered from aorta vein to obtain the plasma and circulating leukocytes for posterior analysis. The lung was recovered and perfused, and the number of B16F10 metastases was counted by visual inspection helped with stereo microscopy (CL 6000 LED, Zeiss, Oberkochen, Germany), followed by fixation.

### 2.7. Histology of Human Biopsies and Mice Lungs

Mice and human biopsies were fixed in 4% buffered paraformaldehyde, dehydrated and embedded in paraffin. Slides sections (4 μm) were dewaxed with xylene and rehydrated through gradient ethanol into water. Hydrated slides were stained with hematoxylin/eosin (H&E EasyPath, São Paulo, Brazil) for morphological analyses or used for immunofluorescence or immunohistochemistry assays.

### 2.8. Immunofluorescence

The immunofluorescence was performed according to Zaqout et al. (2020) [52]. Briefly, for antigen retrieval, sections were heated in citrate buffer (pH 6.0) for 40 min at 95 °C in a water bath. After cooling at room temperature, the sections were rinsed and permeabilized with 0.2% gelatin from cold water fish skin (#G7041 Sigma Aldrich) and 0.25% Triton-X-100 (#X100 Sigma-Aldrich) in PBS. Sections were blocked with 5% bovine serum albumin (BSA; #A7030 Sigma Aldrich), 0.2% gelatin, and 0.25% Triton-X-100 in PBS. Patients’ samples were incubated with anti-AnxA1 mouse antibody (1:50; #610066 BD Biosciences, San Jose, CA, USA) overnight (4 °C), followed by incubation with anti-mouse Alexa Fluor 488 (1:250; #10680 Invitrogen). Lung sections were processed as mentioned previously and incubated with anti-rat Ly6G (1:25; clone 1A8—#16-9668-82 Bioscience/Invitrogen) and anti-AnxA1 rabbit antibody (1:50; #71-3400 Invitrogen). Afterwards, samples were incubated with anti-rat-Alexa-Fluor 488 (1:250; #A1106 Invitrogen) and anti-rabbit Alexa Fluor 568 (1:100; A11001 Invitrogen). The nuclei were stained with DAPI (1 μg/mL; #554907 BD Bioscience). Negative controls were obtained by omitting the primary antibody. Immunofluorescence was analyzed using an Axioskop 2 fluorescence microscope (Carl Zeiss, Jena, Germany), and images were captured using the software AxioVision 4.7 (Carl Zeiss). The Ly6G^+^-positive cell count was performed using the cell counter tool from Fiji/ImageJ software (Version 2.9.0/1.53t National Institutes of Health, Bethesda, MD, USA). Five fields were analyzed from each lung section. The analysis of AnxA1 expression in Ly6G+-positive cells was performed according to de Paula-Silva and collaborators [53]. Briefly, composite pictures (czi format) were imported to Fiji from ImageJ and split into blue and green or red channels. For densitometric analysis, the green or red channel was selected, further analyzed, and modified in the gray filter. Background pixel averages were then subtracted from the image pixels of interest to correct uneven illumination. Fluorescence measures were performed manually by the selection of positive regions; average values were expressed in arbitrary units.

### 2.9. Immunohistochemistry

After antigen retrieval and permeabilization with 0.1% Triton-X-100 in PBS, peroxidase was inactivated with 3% hydrogen peroxide (Synth, Diadema, Brazil), and nonspecific antibody binding was blocked by preincubating sections with 10% BSA in PBS at 37 °C for 60 min. Sections were incubated with diluted primary antibodies against melanoma marker Melan A (1:50; #PA5-99174 Invitrogen) or AnxA1 (1:500, #713400 Thermo Fisher) at 4 °C overnight. Then, slices were washed and incubated with HRP-conjugated anti-rabbit (#ab6721 Abcam, Cambridge, UK) at room temperature for 1 h. The sections were developed with 3,3-diaminobenzidine (DAB; #K3468 Dako, Carpinteria, CA, USA) and lightly counterstained with hematoxylin (Merck, Darmstadt, Germany), followed by dehydration and coverslip mounting. Negative controls were performed by omitting the primary antibody.

### 2.10. Serum and Cell Supernatant AnxA1 Measurement 

The levels of AnxA1 in human and mouse blood and in the neutrophil-conditioned medium were quantified by ELISA using commercial kits (human: #MBS495574; mouse: #M704042, MyBioSource, San Diego, CA, USA) according to supplier instructions.

### 2.11. Flow Cytometry

Flow cytometry experiments were performed to characterize the expression of AnxA1, FPR1, FPR2, CD66b, Ly6G, and F4/80. To evaluate the expression of AnxA1 and CD66b in human neutrophils, isolated cells were fixed with FACS lysing solution (#349202 BD Biosciences), washed with PBS containing 0.1M glycine (Synth), permeabilized with 0.01% Triton-X-100, washed with 1% BSA in PBS (BSA/PBS), and incubated with primary anti-rabbit antibody to AnxA1 (1:100) for 1 h at 37 °C. Next, cells were washed with BSA/PBS and incubated with secondary goat anti-rabbit antibody conjugated to Alexa Fluor 488 (1:250; Invitrogen) and anti-PerCP-Cy™5.5 anti-human CD66b-neutrophil marker (1:50; #562254 BD Pharmigen) for 40 min in the dark at room temperature. To verify the frequency of mice circulating neutrophils (Ly6G^+^) and monocytes (F4/80^+^), cells were fixed with FACS lysing solution for 30 min at room temperature, washed with PBS containing 0.1M glycine, and incubated with anti-Ly6G conjugated with FITC (1:50; #551460 BD Pharmigen) and anti-F4/80 conjugated with PerCP-Cy5.5 (1:200; # 15-4801-82 Biolegend, San Diego, CA, USA) for 40 min in the dark at room temperature. The cells were analyzed by a flow cytometer (BD Accuri C6), taking 10,000 events into consideration and using BD CSampler™ Analysis software (version: 1.0.2641-21 BD Biosciences, Franklin Lakes, NJ, USA). 

### 2.12. Transwell Matrigel Invasion Assay

Transwell membranes (8 μm pore size; #353097 Falcon/Corning) were coated with 40 μL Matrigel (#356237 Corning diluted 1:6 in serum-free DMEM). B16F10 melanoma cells (2.0 × 10^4^ cells/transwell) were resuspended in DMEM supplemented with 0.5% BSA and added to the inner compartment of the chamber [54]. One set of experiments was performed to investigate the role of the neutrophil-conditioned medium (NCM; supernatant from 18 h neutrophils cultured obtained from wild-type or AnxA1^-/-^ mice) on melanoma cell invasion. Hence NCM:DMEM 5% FBS culture medium (300 μL; 1:1 dilution) or DMEM 5% FBS (Control) were added to the outer cup. Another set of experiments investigated the role of the contact of neutrophil–melanoma cells on invasion. In this case, neutrophils (obtained from wild-type or AnxA1^-/-^ mice) and melanoma cells were co-cultured in a proportion of 5:1 for 24 h. The participation of FPRs on the effects was evaluated by treatment of melanoma cells with FPRs antagonists, cyclosporine H (1 μM; #AG CN2 0447-M005 Adipo Gen Life Sciences, San Diego, CA, USA), or WRW4 (1 μM; #2262 Tocris Bioscience, Bristol, UK) for 30 min and before adding to the inner compartment of the chamber. To confirm the involvement of AnxA1, NCM with low amounts of AnxA1 or AnxA1^-/-^ NCM were supplemented with recombinant AnxA1 (1.5 and 3.5 ng/mL; donated by Professor Chris Reutelingsperger from Cardiovascular Research Institute Maastricht, Maastricht University, Maastricht, The Netherlands). After 24 h, cells that migrated through the Matrigel were fixed in 1.0% glutaraldehyde solution (#354400 Sigma Aldrich) and stained with 0.5% toluidine blue in 2% of Na_2_CO_3_ (Synth) for 20 min. The number of migrated cells was counted under a Leica DMi1 inverted microscope (Leica, Shinagawa, Tokyo, Japan). In each well, seven independent fields were considered for quantitation. The data are represented as change fold in relation to control or total number of invading cells. 

### 2.13. Statistical Analysis

Statistical analyses were performed using Graph Pad Prism4 (Graph Pad Software Inc., San Diego, CA, USA). The data were expressed as mean ± standard error of the mean (SEM), and comparisons were made between the experimental groups using the t-test or one-way ANOVA followed by Tukey’s post hoc test for multiple comparisons using GraphPad software version 5 (San Diego, CA, USA). The level of significance was set at *p* < 0.05.

## 3. Results

### 3.1. Expression and Serum AnxA1 Levels Were Increased in Melanoma Patients

The samples included were obtained from nevus or melanoma patients with different Breslow index and stages, or *in situ* melanoma (Table 1). The inflammatory infiltrate was classified as mild to severe, represented mainly by lymphocytes identified by morphological characteristics (Figure 1A–C—black arrows and inserts). AnxA1 expression was verified in four biopsy specimens (dysplastic nevus = 1; melanoma = 3). The selected melanoma samples showed Breslow indexes around 0.8 mm (Figure 2F), 1.1 mm (Figure 2H), and 3.2 mm (Figure 2J). As observed by H&E staining, the samples showed pigmented cells (*) and melanoma cells (black arrow) characterized as large and rounded cells with large nuclei (Figure 1D,F,H,J). In the dysplastic nevus and first melanoma samples, cells scattered throughout the dermis, as seen by H&E staining (Figure 1D,F), were positive for AnxA1 in both samples (Figure 1E,G). In the second melanoma sample, an increased number of pigmented and melanoma cells (Figure 1H) and AnxA1-positive cells (Figure 1I) were detected. In the third sample, the majority of cell types found in the dermis were melanoma cells (Figure 1J), being that almost all of them expressed AnxA1 (Figure K). Peripheral blood was also analyzed. Patients with melanoma showed an increased circulating neutrophil compared to the nevus (Figure 1L). In melanoma patients, the NLR was around 2.5, while in nevus patients, the NLR was 1.69. Furthermore, a high expression of AnxA1 was found in circulating neutrophils (CD66b positive cells) from melanoma patients compared to nevus patients and the control (Figure 1M), and serum AnxA1 levels were increased in patients with melanoma (Figure 1N).

### 3.2. Neutrophils Were AnxA1^+^ in The Lung Melanoma Metastasis Model

On the basis of the expression of AnxA1 in melanoma cells and neutrophils from melanoma patients, a model of lung melanoma metastasis was employed to investigate the pattern of AnxA1 expression and neutrophil infiltration in the metastatic site. As shown in Figure 2A, lung metastasis was observed 21 days after i.v. injection of B16F10. In the lungs of control animals, AnxA1 labeling was detected in the alveolar (arrowheads) but not in the bronchiolar epithelium (Br—Figure 2B,C). Conversely, the lung melanoma metastasis was composed of AnxA1^+^ tumor cells (T—Figure 2D,E) and immune cells (arrows—Figure 2F,G). As verified by Ly6G-neutrophil marker labeling (Figure 2H,I), the number of neutrophils increased in the melanoma group (Figure 2J). Furthermore, Ly6G and AnxA1 double labeling (Figure 2K) or AnxA1 melanoma labeled cells (Figure 2L) showed that neutrophils expressed higher amounts of AnxA1 than melanoma cells (Figure 2M).

### 3.3. Peripheral Blood Neutrophils and AnxA1 Levels Were Enhanced in the Lung Melanoma Metastasis Model

The number of circulating neutrophils was elevated in melanoma metastasis mice in comparison to that found in the control group (Figure 3A). Moreover, levels of AnxA1 in the serum were augmented in lung melanoma metastasis mice (Figure 3B). To verify the ability of neutrophils to secrete AnxA1, peritoneal neutrophils were collected from naive mice, and greater amounts of AnxA1 release were found in the supernatant from these cells than B16F10 cells when cultured for 24 h (Figure 3C). 

### 3.4. AnxA1 Secreted by Neutrophils Increased the Melanoma Cell Invasion via FPRs Pathways 

On the basis of these previous data, we further investigated the involvement of AnxA1 secreted by neutrophils on melanoma cell invasion. Thus, neutrophils were recovered from the peritoneal cavity as depicted in Appendix A. Almost 80% of the total peritoneal cells were neutrophils, and 20% were macrophages as observed by Ly6G and F4/80 labeling, respectively (Appendix A). After macrophage removal by plated-adherence for 2 h, neutrophils were incubated for 18 h, and viability was reduced with an increase in Annexin V, an apoptosis marker (Appendix A), which was expected, since neutrophils’ life span during *in vitro* incubation is around 7 to 10 h for both human and mice cells [55,56]. Furthermore, the levels of AnxA1 secreted by the neutrophil supernatant, named neutrophil-conditioned medium (NCM), was 2 up to 8 ng/mL (Appendix A). Considering that AnxA1 effects are attributed to its cleaved form, NCM was analyzed by Western blotting. As shown in Appendix A, the AnxA1 found in the NCM was cleaved. On the basis of the concentration of AnxA1, NCM was divided into high and low AnxA1 content and employed to investigate the invasion of melanoma cells. For this, murine melanoma B16F10 cells were applied, in which AnxA1 and its receptors (FPR1 and FPR2) were first characterized (Appendix A). B16F10 cells were incubated with NCM, at a proportion of 1:1, with a standard culture medium (DMEM supplemented with 5% FBS) or with standard medium (control) for 24 h (Figure 4A). In the assayed conditions, high AnxA1 content NCM increased the melanoma cell invasion in comparison to the control (Figure 4B), which was not associated with increased cell proliferation (Appendix A). To evaluate the involvement of AnxA1 in the effect promoted by NCM, melanoma cells were pre-treated with cyclosporine (CsH) or WRW4, antagonists for FPR1 and FPR2, respectively. As observed in Figure 4B, incubation with the antagonists reduced the melanoma cell invasion, which was not due to alterations in the cell viability (Appendix A). Further, in NCM containing low levels of AnxA1 (lower than 2 ng/mL), the melanoma cell invasion was reduced in comparison to NCM with detectable AnxA1 levels. The addition of rAnxA1 to non-detectable AnxA1 NCM recovered the cell invasion previously observed (Figure 4C). The role of AnxA1 on cell invasion was corroborated using NCM from neutrophils collected from AnxA1 knockout mice (AnxA1^-/-^). In this condition, the melanoma cell invasion was reduced, which was rescued by the addition of the rAnxA1 (Figure 4D).

The melanoma cell invasion was also evaluated in B16F10-neutrophil co-culture (Figure 4E). As observed in Figure 4F, neutrophil co-culture enhanced the melanoma cell invasion, which was inhibited if melanoma cells were pre-treated with FPR1 and FPR2 antagonists. The reduction of cell invasiveness was also observed if neutrophils were collected from AnxA1^-/-^ mice (Figure 4G).

We also investigated whether melanoma-conditioned medium (MCM) could change the neutrophil’s phenotype. Indeed, MCM reduced the expression of ICAM-1 at the neutrophil’s surface (Appendix A) and increased the secretion of CXCCL1 KC, MCP-1, Arginase, VEGF, and IL-10 (Appendix A). As this neutrophil’s profile is characteristic of the N2 phenotype, it may indicate that neutrophils at the site of metastasis display an anti-inflammatory phenotype.

### 3.5. Depletion of Circulating Neutrophils Reduced Lung Melanoma Metastasis and Serum Levels of AnxA1

Once AnxA1 secreted by neutrophils displayed a role in melanoma metastasis, we further investigated whether depletion of these cells during the disease progression was able to alter the disease outcomes. We depleted circulating neutrophils using anti-granulocyte antibody anti-Gr1 before B16F10 injection until the end of the lung melanoma metastasis model (Figure 5A). In our model, the antibody dosage and schedule reduced the number of peripheral neutrophils during the 48 h period after the antibody injection (Figure 5a). There was no detected modification in the number of circulating monocytes in the blood, while the number of blood neutrophils remained reduced up to day 21 (Appendix A). The remaining neutrophils showed morphology like an immature phenotype (Appendix A), which may have been the result of bone marrow response to anti-rat antibody production (mitigation), due to limited depletion in peripheral tissue (low bioavailability), or due to the induction of neutrophil production in the spleen (extramedullary granulopoiesis), as discussed by recent literature [57,58].

Even the completed neutrophil depletion was not achieved; we observed a significant reduction of melanoma metastasis spots as observed by macroscopic analysis of the lungs from depleted neutrophils (anti-Gr1 treaded) in comparison to the isotype group (Figure 5B–D). Histologic analysis using H&E staining showed that the isotype group had metastasis in the external regions of lobes and near blood vessels (Figure 5E). The lungs of neutrophil-depleted mice showed tumor mass spots near vessels; however, the size was smaller than that found in the isotype group (Figure 5F). Further, we observed an increase in the circulating lymphocytes in the neutrophils depleted animals in comparison to isotype mice (Appendix A). Bronchoalveolar lavage (BAL) showed a reduced level of TGF-ß and VEFG-α (Appendix A), which may have been related to the reduced number of neutrophils in the lungs. To characterize lungs from neutrophil-depleted mice, we performed immunohistochemistry analysis using a melanoma marker (Melan A). Thus, it was verified that the Melan A-positive cells were restricted to the tumor (Figure 5G). On the other hand, in neutrophil-depleted mice, Melan A-positive cells were found to be mixed with other Melan A non-labeled cells also near vessels (Figure 5H). It was also observed that Melan A-positive cells were found spreading in the lung tissue of depleted-neutrophil mice (Figure 5H—insert). The AnxA1 expression was found in melanoma tumor cells (Figure 5I,J) and immune cells (Figure 5i,j) in the lung from both isotype and neutrophil-depleted mice. Furthermore, the depletion of neutrophils reduced the levels of serum AnxA1 (Figure 5E), reinforcing the potential of AnxA1 as a melanoma biomarker.

## 4. Discussion

Neutrophils have been recognized as relevant players in cancer biology. Their actions are under robust investigation as they display anti- or pro-tumoral activities. The duality of neutrophil behavior is dependent on the phase of the disease, tumor environment contents, and the state of neutrophil activation in the blood or tumor sites. AnxA1^+^ neutrophils are fundamental to halting inflammation, and although AnxA1 secreted by cancer cells is involved in metastasis development, the role of AnxA1 derived from neutrophils in tumor invasiveness is underestimated. Hence, we addressed this issue, showing AnxA1 secreted by neutrophils contributes to the development of melanoma lung metastasis by promoting cancer invasiveness through FPR1/FPR2 pathways.

The up-expression of AnxA1 by tumor cells takes place in solid tumors of different origins and is associated with poor outcomes [59,60,61,62], implicating AnxA1 as a common mediator of cancer development. Indeed, aggressive human melanoma cell lines express high amounts of AnxA1 related to cell invasiveness, as verified by impairing melanoma cell invasion when endogenous AnxA1 protein levels were reduced [46]. Moreover, the treatment of the melanoma cells with the mimetic AnxA1 N-terminal peptide Ac2–26 stimulated the invasiveness by increasing MMP-2 expression, depending on FPR/MAPK/STAT3 activation pathways [46,47,48]. Furthermore, impaired melanoma lung metastasis in AnxA1^-/-^ mice was associated with the inhibition of angiogenesis [63], pointing out that AnxA1 is a player protein such as on cancer as stromal cells. A previous retrospective clinical study showed that high AnxA1 expression in melanoma cells from primary tumors reduced metastasis-free survival, but AnxA1 expression levels were unchanged according to the Breslow index [46]. Here, data obtained in melanoma patients corroborate this evidence, as augmented AnxA1 expression in the cytoplasm of cells from a dysplastic nevus and melanoma patients was detected. However, the intensity of AnxA1 labeling did not change according to melanoma cells migrating deeper or spreading into the dermis, suggesting that maintenance of constitutive AnxA1 expression is an effector molecule mediating the tumor progression.

The NLR is a readily available metric with an emerging role in melanoma prognosis [15]. For localized melanoma, a high NLR is predictive of worse overall survival. However, the optimal cut-off for NLR is not established, varying between 2 and 5 [15]. Patients with NLR > 5 had significantly worse median overall survival in comparison with low NLR [64]. Similarly, NLR value has been considered in the prognostic of the patients receiving checkpoint inhibitors [24,25,26]. For instance, Bartlett et al. (2020) demonstrated that patients from the high-NLR group were more likely to have higher disease burden and poorer overall performance status. Furthermore, an NLR increase of ≥30% after initiation of therapy was also associated with shortened overall survival [65]. Although the number of patients included in our study was small, we observed a peripheral neutrophilia and the NLR was around 2.5, which is within the range considered for localized melanoma. Interestingly, we also observed that blood neutrophils from melanoma samples showed increased AnxA1 expression compared to nevus. The function of circulating AnxA1^+^ neutrophils in melanoma is not yet known and deserves further investigation. Nonetheless, AnxA1^+^ neutrophils in the blood are related to an immunosuppressive profile [39,41,66,67]. Furthermore, the potential of AnxA1 serum levels as a tumor biomarker has not been well explored. Its upregulation was only associated with pathological grade and clinical stage in patients with lung cancer [68]. Here, high serum AnxA1 levels and neutrophils expressing AnxA1 were consistent, such as that found in melanoma samples from patients as in the melanoma mice model. The fact that we found an increase in AnxA1 serum levels brings an alternative approach to help in the diagnosis of melanoma, and a large cohort is needed to support the potential of AnxA1 as a biomarker for melanoma detection or progression.

Further experimental data showed that neutrophils that infiltrated into the melanoma lung metastasis expressed elevated levels of AnxA1 in comparison to melanoma cells. A broad amount of data show that AnxA1 secreted by neutrophils into the inflammatory environment is phosphorylated and exerts autocrine and paracrine actions to halt the inflammatory process and to induce efferocytosis [69]. To the best of our knowledge, we suggest, for the first time, that neutrophils secrete AnxA1 in melanoma metastasis and can influence tumor development. Hence, to confirm this hypothesis, an *in vitro* platform was carried out to detect the role of neutrophils or neutrophil-secreted products (NCM) on tumor cell invasiveness. Indeed, such melanoma cells co-cultured with neutrophils as incubated with NCM presented higher invasiveness, which was rescued if neutrophils were obtained from AnxA1^-/-^ mice. The antagonisms of FPR1 or FPR2 also inhibited the augmented invasiveness of melanoma cells co-cultured with neutrophils or incubated with NCM. In the conditions where rAnxA1 was added to the medium, the melanoma cell invasion was lower than that found in NCM. Studies have shown that the active form of AnxA1 is associated with the N-terminal region, which can be exposed after a conformational change at the neutrophil plasma membrane or cleaved in a 33 kDa fragment by enzymes as elastase [70,71]. Here, we verified that AnxA1 secreted by neutrophils is cleaved. The lower effect promoted by rAnxA1 in comparison to NCM could be associated with the lack of the cleavage process. Hence, this body of data drives a pivotal role of AnxA1 secreted by neutrophils in melanoma invasion via the FPRs axis.

We here confirmed that B16F10 cells constitutively express FPR1 and FPR2, and data from the literature show that AnxA1 binds to both receptors and leads to cancer progression, such as breast cancer [67]. Interaction of AnxA1 with FPRs modulates signal transduction pathways, such as oncogenic signaling and extracellular-signal-related kinase (ERK) phosphorylation, resulting in cell invasion [72]. Indeed, the MAP kinase pathway is associated with melanoma development, as the inhibition of the members of the RAS–RAF–MEK–ERK axis has been a target of recent drugs to treat melanoma [73]. Moreover, previous studies showed that AnxA1 stimulated MMP-2 activity by interaction with FPRs, increasing cell invasiveness and promoting the proliferation and migration of melanoma cells *in vitro* [48,74].

Nonetheless, we may also suggest that AnxA1 secreted by N2 phenotype neutrophils influences stromal and immune cell behavior in the microenvironment, favoring the tumor progress. Indeed, AnxA1 via FPR1 potentiates the VEGF-A signaling in endothelial cells by promoting new vessel formation [75], and neutrophil-derived AnxA1 is related to angiogenesis during the development of the placenta [43]. Additionally, the reduction of lung melanoma metastasis in AnxA1^-/-^ mice was associated with neovascularization impairment [63]. As mentioned before, N2 neutrophil-secreting AnxA1 displays immunoregulatory effects [39,41,66,67], which may effectively contribute to the immunosuppressive status of the metastasis sites. Robust evidences show a complex interplay of chemical mediators secreted by cancer cells and neutrophils on tumor progression when mediators secreted by cancer cells induce the switch of neutrophils to the N2 phenotype [75,76]. A recent report demonstrated that melanoma cell–neutrophil interaction supports cancer progression by priming neutrophils towards a pro-tumor N2 phenotype [77]. Indeed, the data here obtained show that melanoma-conditioned medium polarizes neutrophils into the N2 phenotype. 

In addition, we observed a reduction of melanoma metastasis formation and serum levels of AnxA1 in neutrophil-depleted mice. Furthermore, an increase in lymphocytes was detected in lung metastasis, indicating that neutrophils at the tumor site may act as immunosuppressive cells. Indeed, neutrophil depletion has unveiled its role on cancer progression by suppressing antitumor T cells [78]. T-cell activation can be inhibited directly by TGF-β or IL-10 secreted by N2 phenotype [19] or indirectly by MMP-9 secretion that activates the TGF-β by proteolytic cleavage [79]. Moreover, neutrophils favor melanoma dissemination by releasing the neutrophil extracellular traps (NETs) induced by complement membrane attack complexes; NETs lead to the endothelial barrier opening that allows melanoma cells to enter the circulation and systemic spread [29]. Indeed, lung metastasis induced by melanoma cells expressing AnxA1 is impaired in AnxA1^-/-^ mice [63], addressing the fact that constitutive AnxA1 secreted by other mice cells are involved in the metastasis. Herein, the reduction of neutrophils implied in the serum AnxA1 secretion decrease, which may impair melanoma cell invasion and, consequently, its dissemination.

Our findings added AnxA1 secreted by neutrophils in the blood or at metastasis sites as a new player of the melanoma cell invasion, pointing to AnxA1 as a pivotal mediator secreted by neutrophils acting on cancer cells. These data open a venue for investigations about the mechanisms of AnxA1 secreted by neutrophils, such as in the blood as in tumor metastasis and to propose AnxA1 blood levels or AnxA1^+^ neutrophils as a biomarker of early detection or melanoma progression. 

## Figures and Tables

**Figure 1 cells-12-00425-f001:**
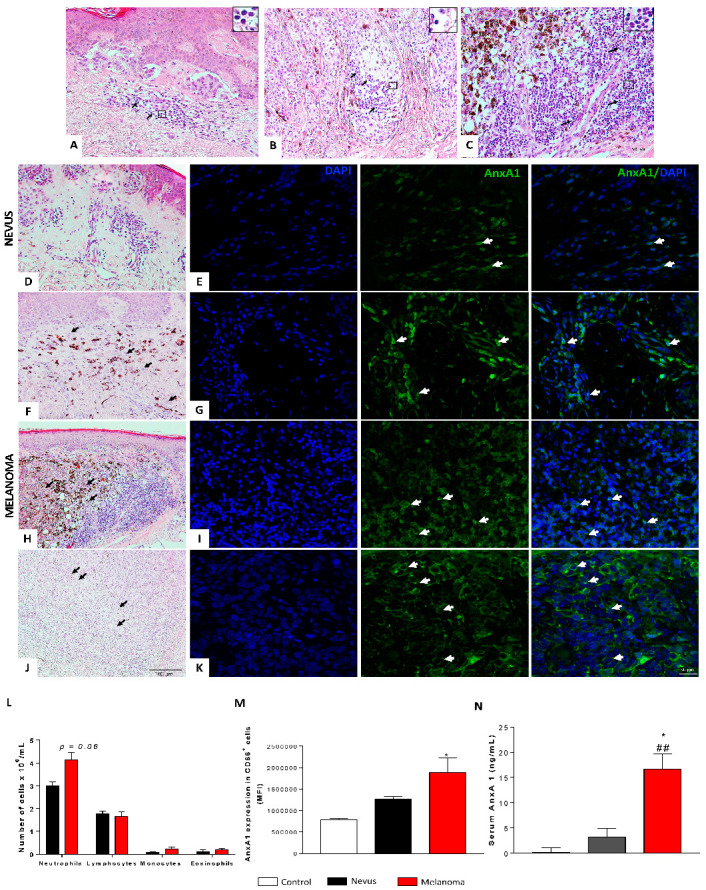
Expression and serum levels of AnxA1 were increased in melanoma patient samples. (**A**–**C**) Representative image of inflammatory infiltrate in biopsies obtained from melanoma patients—H&E staining (black arrows and inserts: lymphocytes). Scale bar: 100 µm. (**D**) H&E staining and (**E**) AnxA1 expression detected by immunofluorescence in nevus sample. (**F**,**H**,**J**) H&E staining and (**G**,**I**,**K**) AnxA1 expression detected by immunofluorescence in melanoma cells from melanoma biopsies. (*) pigmented cells. (Black arrows) melanoma cells. (White arrows) AnxA1-labeled cells. Scale bar: 50 µm. (**L**) Leukogram. (**M**) AnxA1 expression in circulating neutrophils (CD66^+^) evaluated by flow cytometer. (**N**) AnxA1 serum levels. The data represent the average ± SEM. Control (*n* = 4); nevus (*n* = 4–5); melanoma (*n* = 5–9). * *p* < 0.05 vs. nevus; ## *p* < 0.01 vs. control.

**Figure 2 cells-12-00425-f002:**
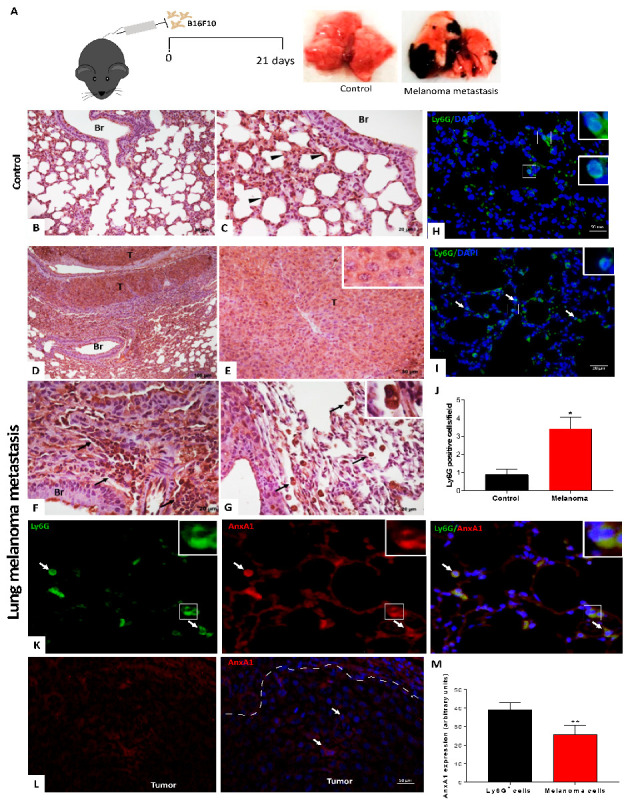
Increased expression and serum levels of AnxA1 in lung melanoma metastasis mice. (**A**) Experimental design. (**B**,**C**) Control group. Alveolar epithelium (arrowhead). Bronchiolar epithelium (Br). (**D**,**E**) Expression of AnxA1 in melanoma cells (T) from lung melanoma metastasis. (**F**,**G**) Immune cells labeled for AnxA1. Immune cells infiltrate (arrows). Inserts: Tumor and immune cells. Ly6G-positive cells in (**H**) control and (**I**) melanoma groups. (**J**) Increased number of neutrophils in lung melanoma metastasis mice. (**K**) AnxA1 expression in neutrophils (Ly6G^+^) and (**L**) in melanoma cells from lung melanoma metastasis. (**M**) Amounts of AnxA1 expression in neutrophils and melanoma cells. The data represent the average ± SEM of at least 3–5 animals per group. *p* * < 0.05 vs. control; ** *p* < 0.01 vs. neutrophils.

**Figure 3 cells-12-00425-f003:**
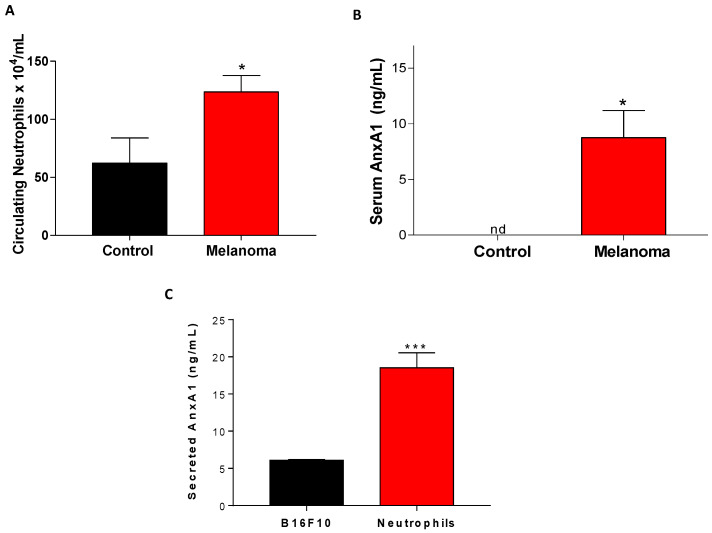
Increased blood neutrophils and AnxA1 levels in the lung melanoma metastasis model. (**A**) Amounts of neutrophils in the peripheral blood from lung melanoma metastasis mice. (**B**) Serum levels of AnxA1. (**C**) Secretion of AnxA1 by melanoma cells (B16F10) and neutrophils cultured for 24 h. The data represent the average ± SEM. * *p* < 0.05 vs. control; *** *p* < 0.001 vs. B16F10. Control group (*n* = 5); melanoma group (*n* = 5).

**Figure 4 cells-12-00425-f004:**
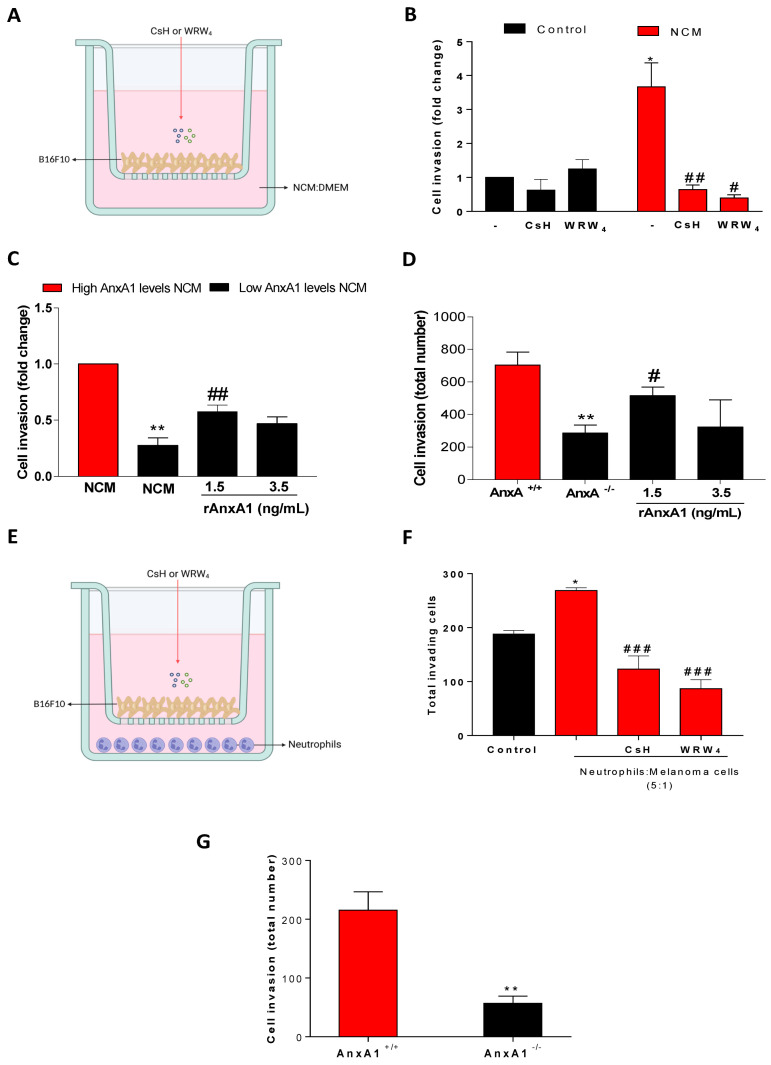
Neutrophil-conditioned medium or neutrophil cultured with melanoma cells increased the cell invasion by in a dependent manner of AnxA1. (**A**) Experimental design. (**B**) Invasion of melanoma cells in the absence and presence of FPR inhibitors incubated with standard culture medium (DMEM supplemented with 5% of FBS–control) or neutrophil-conditioned medium (NCM) for 24 h. Melanoma cells were pre-treated with 1 μM cyclosporine H (CsH; FPR1 inhibitor) or 1 μM WRW_4_ (FPR2; inhibitor) for 15 min before seeding at the transwell bottom. * *p* < 0.05 vs. control; # *p* < 0.05; ## *p* < 0.001 vs. NCM. (**C**) Invasion of melanoma cells in the presence of NCM—detectable levels of AnxA1, NCM—undetectable levels of AnxA1, or NCM—undetectable levels of AnxA1 supplemented with recombinant AnxA1 (rAnxA1) at indicated concentrations. **/## *p* < 0.01 vs. NCM. (**D**) Invasion of melanoma cells in the presence of NCM from wild-type (AnxA1^+/+^) or AnxA1 knockout (AnxA1^-/-^) mice cultured for 24 h. (**E**) Experimental design. (**F**) Invasion of melanoma cells in the absence and presence of FPR inhibitors co-cultured for 24 h with neutrophils at proportion of 5 neutrophils to 1 melanoma cell. * *p* < 0.05 vs. control; ### *p* < 0.001 vs. untreated cells. (**G**) Invasion of melanoma cells in the presence of neutrophils from AnxA1^+/+^ or AnxA1^-/-^ mice cultured for 24 h. The data represent the average ± SEM of at least five to eight independent experiments. ** *p* < 0.01 vs. AnxA1^+/+^.

**Figure 5 cells-12-00425-f005:**
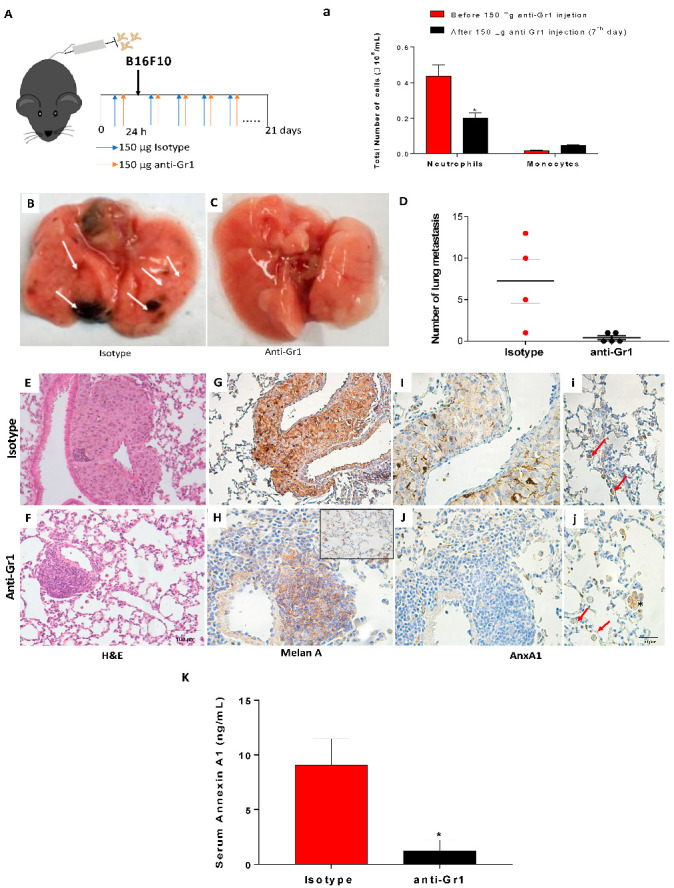
Neutrophil depletion reduced the number of melanoma lung metastasis and serum AnxA1 levels. (**A**) Experimental design. (**a**) Total number of neutrophils and monocytes in peripheral blood. (**B**,**C**) Images of the lungs from isotype or anti-Gr1-treated mice. (**D**) Number of melanoma metastasis spots. (**E**,**F**) Morphologic analysis by H&E staining. (**G**,**H**) Representative images of the Melan A expression (**I**,**i**) in isotype and (**J**,**j**) neutrophil-depleted mice. Insert: Melan A expressed in cells spread into the lungs. Red arrows: immune cells labeled for AnxA1. (**K**) Serum levels of AnxA1. The data represented the average ± SEM. * *p* < 0.05 vs. isotype. Melanoma group (*n* = 4, 5). Melanoma neutrophil-depleted group (*n* = 5).

**Table 1 cells-12-00425-t001:** Characteristics of patient’s sample. ***: absence. SLN: sentinel lymph node.

Characteristics								
Total Patients Enrolled (*n*) = 16								
Age (Years) (*Mean ± SD*) = 55.93 ± 15.25								
Patients	Skin Phototype	Site	Diagnostic	Breslow Thickness (mm)	Ulceration	Inflammatory Infiltrate	SLN	Stage
**1**	I	Left infra scapular	Melanoma	0.6	No	Slight	No	IA
**2**	NA	Left abdomen	Melanoma	14	No	Discrete	Yes	IV
**3**	NA	Right leg	Melanoma	0.6	No	Mild	No	IA
**4**	II	Right upper back	Melanoma	3.2	No	Discrete	No	IIA
**5**	II	Left flank	Compound nevus	0	No	Discrete	No	0
**6**	III	Right scapular	Melanoma (lentigo maligno)	0	No	0	No	0
**7**	III	Left inframammary	Compound nevus	0	No	***	No	0
**8**	-	Left thigh	Residual melanoma (lentigo maligno)	0	No	0	No	IB
**9**	III	Right thorax (breast)	Atypical nevus	0	No	0	No	0
**10**	II	Left lumbar	Melanoma	0.8	No	Discrete	No	IIIA
**11**	III	Right arm	Melanoma	2.5	No	Discrete	No	IIA
**12**	II	Right thoracolumbar	Melanoma (ES)	1.1	No	Slight	No	0
**13**	III	Left scapular	Melanoma	*in situ*	No	Intense	No	0
**14**	II	Right scapular	Dysplastic melanocytic nevus	0	No	***	No	0
**15**	II	Right lower abdomen	Dysplastic melanocytic nevus	0	No	***	No	0
**16**	II	Right axillary	Dysplastic junctional melanocytic nevus	0	No	***	No	0

## Data Availability

Not applicable.

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
