# Peer review of "Role of Annexin A1 Secreted by Neutrophils in Melanoma Metastasis"

_cells, 2023, doi:10.3390/cells12030425_

Round 1

Reviewer 1 Report

In this study, S. Sandri et.al. investigated the role of AnxA1 secreted by neutrophils in melanoma metastasis. This study proposed the AnxA1 blood levels or AnxA1+ neutrophils as a biomarker for melanoma progression. The study presents interesting results, but unfortunately, the data are not solid enough to prove their hypothesis and some of the mechanisms proposed require further explanation or experimentation:

1, In figure1, the authors use human melanoma samples to investigate the expression and serum level of AnxA1. Are neutrophils in human primary melanoma samples (figure 1G,I,K,) also higher AnxA1 expression compared with melanoma cells? Figure 1M and 1N show that neutrophils in the control group still express AnxA1 (almost 40% compared with the melanoma group) but why the serum level of AnaxA1 in the control group are almost non-detectable? What’s the mechanism for the regulation of neutrophil AnxA1 release? Figure 1N show that serum AnxA1 level is increased in melanoma patients, please indicate the melanoma blood samples stage. Is there any correlation between the stage and serum level & neutrophil surface expression levels of AnxA1? Any correlation between the AnxA1 level and melanoma patients' survival?

2, In figure2, the authors use i.v. injection lung melanoma metastasis tissue to prove neutrophils are AnxA1 positive, however, the staining quality is not so good, especially the AnxA1 staining (looks like unspecific staining, figure 2K, I). Please use a positive control to validate the AnxA1 antibody staining. Figure 2F and 2G are not convincing for the neutrophil and macrophage AnxA1+ staining in lung tissue. Please use immunofluorescence staining (double staining) to further confirm it.

3, Figure 3A shows only 100 neutrophils/ml in melanoma mouse blood, is this right? Figure 3C shows the AnxA1 release from neutrophils after 24 hours of culture. Normally isolated neutrophils died after 24 hours, a time curve for the AnxA1 release will be better. In figure 3c, the authors want to compare the AnxA1 release between melanoma cells B16F10 and neutrophils, a tissue staining will be more convincing. Please check your reference 61 (https://doi.org/10.1073/pnas.0901324106), in this PNAS paper, the author reported that B16 melanoma cells do not express AnxA1 (FigS1A-B).

4, The authors should investigate whether melanoma conditional medium could stimulate neutrophil release AnxA1.

5, Neutrophil depletion experiment cannot support the role of AnxA1 secreted by neutrophils mediating melanoma metastasis.  i.V. melanoma (AnxA1-KO or AnxA1 non-expressed melanoma cell) injection in WT and AnxA1-/- mice experiment should be performed.  If possible, the FPR inhibitor treatment mice experiment should be added.

6, minor point: please confirm your reference 29, line68-71.

Author Response

Dear Reviewer #1, thank you very much for your consideration. All of them were answered below and, when needed, the changes were done using the Track Changes in the manuscript. The answers to the question and your considerations are found in the attached file.

Reviewer 2 Report

This is an interesting study on the role of annexin A1 in melanoma. Here, Annexin A1 was shown to be secreted not only by tumor cells but also by neutrophils in the tumor microenvironment and its impact on the tumor was revealed.

The role of Annexin A1 was already studied in melanoma and several studies (which are discussed in the manuscript) already revealed several functional aspects. Commonly the focus was the expression in the melanoma cells. Further studies already illustrated the role of Annexin A1 in neutrophils e.g. showing that Annexin A1 regulates maturation, recruitment, apoptosis and recirculation. Also, these studies are cited. Other studies concentrated on showing an important role of neutrophils in tumor metastasis including melanoma. Several factors were attributed to these mechanisms like HMGB1 or TNF. This aspect (e.g. potential interplay of Annexin A1 with other known factors) needs to be addressed and discussed in more detail to understand the role of Annexin A1 in this complex context better.   

In summary, the study is mainly reproducing already available knowledge however giving some new insights. 

Major comments:

-       It would be interesting to confirm effects, which are central for the manuscript in more cell lines, best in human melanoma cells to prove that the findings on the mechanisms are valid. The part on human tissue samples is supporting this but further data is necessary. 

-       Figure 1: It is unclear how the cells marked by arrows were identified, e.g. lymphocytes were marked by red arrows but no markers were used. Is the same is true for melanoma cells, here pigmentation was used as marker although it is known that the pigment can be distributed or taken up by other cells. To clarify the nature of the cells, specific staining is needed.

-       Figure 1L: I can not find the description how the measurement was performed.

-       Figure 1N: as the serum level are already elevated in the nevus group, a discussion of its value as a serum marker is needed. Right now, the value seems very limited. Is the level in the blood associated to stage or tumor mass?

-       Figure 2: the quality of the pictures needs to be improved (e.g. 2A and immunohistochemistry). 

-       Figure 2M: How were the tumor cells stained? To really define differences in Annexin A1 level comparing melanoma cells and neutrophils, the analysis has to be performed in one slide using three markers. It further is unclear what tissue was used in 2K.

-       Figure 3: Is the serum level of Annexin A1 correlated in the mouse model to amount of metastasis? 

-       Figure 4: was the effect of annexin A1 on invasion dose-dependent?

-       Figure 4C,D: the effect of recombinant Annexin A1 is always smaller than the conditioned media. Do the authors speculate that additional factors are relevant (see also the general question above)

-       Figure 5: depletion of neutrophils is confirming several previous reports that these cells are important but is not supporting that AnnexinA1 is necessary for their effects. Here, an addition of Annexin A1 in the depleted mice would be the interesting experiment showing a direct effect of Annexin A1 even without neutrophils. This would also answer further question whether only Annexin A1 is sufficient or more factors secreted by neutrophils ae necessary. In the discussion, the interpretation of this experiment needs to be tuned down, as an direct evaluation of Annexin A1 effects is not feasible. 

-       Only one blot is shown in uncropped blots and it is not illustrated in detail what is shown. It further seems that only 2 repetitions are presented? No original data of e.g. the FACS analysis is presented. 

-       Further, no excel files showing the calculation and statistics of the figures are given.

Minor comment: English proofreading is necessary (e.g. “To better of our knowledge,…”)

Author Response

Dear Reviewer #2, thank you very much for your consideration. All of them were answered below and, when needed, the changes done using the Track Changes in the manuscript. The answers to the question and your considerations are found in the attached file.

Round 2

Reviewer 1 Report

In figure 5, the authors used Anti-Gr1 antibody to deplete neutrophils in the mouse blood. Anti-Gr1 is a rat IgG2b that induces neutrophil cell death through complement effect (PMID: 32488020) and it can induce strong inflammation (complement membrane attack complex formation) on neutrophils. The secretion of AnxA1 occurs in inflammatory conditions or by apoptotic neutrophils (doi: 10.1002/emmm.201000113). According to the theory above, anti-Gr1 antibody treatment (neutrophil inflammation and cell death--stronger condition than apoptosis) can induce more neutrophil Annexin A1 secretion into the circulation. In figure5K, the author found less serum levels of Annexin A1 in anti-Gr1 antibody treatment group. Please add a control experiment to prove that anti-Gr1 antibody-induced neutrophil cell death does not lead to Annexin A1 secretion.

Author Response

Dear Reviewer #1, 

The answers are in the attached file.

Reviewer 2 Report

Most points were addressed and resulted in valid changes in the manuscript. Some points were only answered in the rebuttal letter, however, this reviewer feels that more aspects needs to be addressed or explained in the manuscript (e.g. effect of recombinant Annexin A1, or depletion of neutrophils). Here, I can not find changes in the manuscript taking up the comments.

Author Response

Dear Reviewer #2, 

The answers are in the attached file.
